# Cytochrome *c* Oxidase at Full Thrust: Regulation and Biological Consequences to Flying Insects

**DOI:** 10.3390/cells10020470

**Published:** 2021-02-22

**Authors:** Rafael D. Mesquita, Alessandro Gaviraghi, Renata L.S. Gonçalves, Marcos A. Vannier-Santos, Julio A. Mignaco, Carlos Frederico L. Fontes, Luciana E.S.F. Machado, Marcus F. Oliveira

**Affiliations:** 1Departamento de Bioquímica, Instituto de Química, Universidade Federal do Rio de Janeiro, Rio de Janeiro, RJ 21941-909, Brazil; rdmesquita@iq.ufrj.br (R.D.M.); alessandro.gaviraghi@gmail.com (A.G.); 2Instituto Nacional de Ciência e Tecnologia em Entomologia Molecular (INCT-EM), Rio de Janeiro, RJ 21941-590, Brazil; 3Instituto de Bioquímica Médica Leopoldo de Meis, Universidade Federal do Rio de Janeiro, Cidade Universitária, Rio de Janeiro, RJ 21941-590, Brazil; jmignaco@bioqmed.ufrj.br (J.A.M.); cfontes@bioqmed.ufrj.br (C.F.L.F.); 4Sabri Ülker Center for Metabolic Research and Department of Molecular Metabolism, Harvard T.H. Chan School of Public Health, Boston, MA 02115, USA; rgoncal@hsph.harvard.edu; 5Laboratório de Inovações em Terapias, Ensino e Bioprodutos, Instituto Oswaldo Cruz, Fundação Oswaldo Cruz, Fiocruz, Rio de Janeiro, RJ 21040-900, Brazil; marcos.vannier@ioc.fiocruz.br; 6Departamento de Bioquímica, Instituto de Química, Universidade de São Paulo, São Paulo, SP 05508-000, Brazil; lucianaelenasfm@gmail.com; 7Departamento de Genética e Biologia Evolutiva, Instituto de Biociências, Universidade de São Paulo, São Paulo, SP 05508-090, Brazil

**Keywords:** metabolism, bioenergetics, respiration, oxidative phosphorylation, redox, dispersal, migration, homeostasis, kinetics, allostery, phosphomimetic

## Abstract

Flight dispersal represents a key aspect of the evolutionary and ecological success of insects, allowing escape from predators, mating, and colonization of new niches. The huge energy demand posed by flight activity is essentially met by oxidative phosphorylation (OXPHOS) in flight muscle mitochondria. In insects, mitochondrial ATP supply and oxidant production are regulated by several factors, including the energy demand exerted by changes in adenylate balance. Indeed, adenylate directly regulates OXPHOS by targeting both chemiosmotic ATP production and the activities of specific mitochondrial enzymes. In several organisms, cytochrome *c* oxidase (COX) is regulated at transcriptional, post-translational, and allosteric levels, impacting mitochondrial energy metabolism, and redox balance. This review will present the concepts on how COX function contributes to flying insect biology, focusing on the existing examples in the literature where its structure and activity are regulated not only by physiological and environmental factors but also how changes in its activity impacts insect biology. We also performed in silico sequence analyses and determined the structure models of three COX subunits (IV, VIa, and VIc) from different insect species to compare with mammalian orthologs. We observed that the sequences and structure models of COXIV, COXVIa, and COXVIc were quite similar to their mammalian counterparts. Remarkably, specific substitutions to phosphomimetic amino acids at critical phosphorylation sites emerge as hallmarks on insect COX sequences, suggesting a new regulatory mechanism of COX activity. Therefore, by providing a physiological and bioenergetic framework of COX regulation in such metabolically extreme models, we hope to expand the knowledge of this critical enzyme complex and the potential consequences for insect dispersal.

## 1. Introduction

### 1.1. Flight as an Extreme Metabolic Process

Insects are the largest and most diverse animal group in nature, with estimates reaching around 5.5 million species [1]. Insects’ extreme evolutionary and ecological success in dominating nearly all niches of our planet results from their remarkable adaptive abilities. The flight is a key aspect that insects mastered, which brought enormous advantages relative to their wingless counterparts by providing efficient ways to acquire food resources, to escape predators, and to establish in new places. Indeed, flight is not an exclusive biological trait of insects but is also shared with birds and bats (and with extinct pterosaurs), which evolved from four different wingless ancestors [2]. Undoubtedly, the convergent evolution towards flight in four independent animal groups underscores the great adaptive advantage of these organisms to improve survival.

Insects were the first animals to evolve flight, and the most ancient register of a winged insect fossil dates back to circa 324 million years ago (Mya) during the late Paleozoic [3,4,5,6]. Indeed, several fossil registers of giant insect species temporarily match with the first flying insects, particularly the ancient dragonflies of the *Meganeura* genus that reached wingspans of >70 cm [4]. Although the origins of flight are controversial, an interesting hypothesis suggests that fluctuations in atmospheric concentrations of oxygen and carbon dioxide were vital contributors to allow the conditions for the evolution of flight [3,5]. The increase in atmospheric oxygen concentration (apO_2_) may have physiologically facilitated the development of oxidative metabolism, which may have represented the main physiological adaptation necessary to satisfy the energy demand required by the flight activity. Curiously, the surge of giant insects was parallel to the highest apO_2_ ever registered, reaching ≈30% in the late Paleozoic, suggesting an association between oxygen availability and insect’s body size [3]. This assumption was based on the fact that in insects, oxygen is transported through a complex network of tracheae that deliver oxygen directly to the insect’s organs, independent of a circulatory system (Figure 1A). Therefore, higher apO_2_ would circumvent the diffusion limits and allow the development of larger body sizes [7]. Although attractive, this proposal has several limitations as it does not consider, for example, that oxygen supply is morphologically and physiologically adjusted to oxygen demand among species with different body sizes [8,9].

The enormous physical forces required to sustain flight posed several biomechanical, physiological, and metabolic constraints for flying insects [10]. Regardless of their intrinsic anatomical and structural diversities, insect flight muscles share a remarkable feature to allow extremely high wing beat frequencies (in some cases even higher than 800 Hz [11]). Like vertebrate skeletal muscles, insect flight muscle is a cross-striated muscle, and is essentially composed of four components, (*i*) the myofibrils, which are responsible for the work produced during contraction, (*ii*) the sarcoplasmic reticulum, which is involved in the impulse of contraction and subsequent relaxation through the release and sequestration of calcium (Ca^2+^), (*iii*) the mitochondria which are responsible for ATP production to meet the cellular energy demands [12,13,14,15,16,17] and also to produce oxidants including superoxide and hydrogen peroxide [18,19,20,21], and (*iv*) the tracheae that deliver atmospheric oxygen directly to mitochondria. The regulatory mechanism for flight muscle contraction and relaxation is similar to that of vertebrate skeletal muscle: the excitation of muscle plasma membrane induces the release of Ca^2+^ ions from the sarcoplasmic reticulum that bind to a thin filament regulatory protein called troponin. As a consequence, tropomyosin then moves from its inhibitory position to initiate contraction. Conversely, relaxation occurs when Ca^2+^ is pumped back to the sarcoplasmic reticulum lumen. It is important to emphasize that this transport is an energy-consuming active process because Ca^2+^ ions must be pumped against the concentration gradient [22].

Despite the huge respiratory rates observed in flying insects, muscle efficiency (power output generated by mechanical contraction divided by power input resulted from energy metabolism) is much lower in insects than in birds [10,23,24]. Indeed, allometric scaling of metabolic and mechanical power indicates that muscle efficiency increases with body size. In humans, the efficiency of skeletal muscle contractile activity is ≈70%, while for most insects is just 10% [25]. Interestingly, the opposite trend is observed in their thermogenic capacity, which reduces with increasing body size. Many flying insect species are endothermic homeotherms, contrasting with runner insects that are ectothermic poikilotherms [10]. These patterns imply that while flying insects exhibit very high respiratory rates, a significant part of the redox energy from respiration dissipates as heat. Therefore, to compensate for this reduced muscle efficiency, flying insects increase their capacity to extract energy from nutrients oxidation in mitochondria. An interesting observation of the critical thermogenic role of respiration in a flying insect is exemplified in bumblebees. Instead of conserving the energy from nutrient oxidation as ATP by oxidative phosphorylation (OXPHOS), bumblebees mediate thermogenesis through a metabolic shift towards glycerol 3-phosphate (G3P) oxidation during cold weather [26]. Importantly, as G3P oxidation is intrinsically less coupled to ATP production than other substrates [19,21,26,27], preflight thermogenesis is thus facilitated, allowing flight muscles to warm in cold weather [26].

Insect flight muscles can be anatomically divided into two distinct groups: the direct flight muscles (DFM), which connect the thoracic cuticle to wings bases and drive the downward movement of wings, and the indirect flight muscles (IFM) that drive the upward movement of wings. Flight muscles can also be physiologically divided into synchronous flight muscles (SFM) and asynchronous flight muscles (AFM), depending on the relationship between neural electrical stimulation and muscle mechanical work [28,29]. While SFM has a single contraction upon a neural stimulus, AFM contracts multiple times in an oscillatory manner, usually at high frequencies. SFM is considered the most primitive and is characteristic of some species of locusts, butterflies, and moths [12]. Interestingly, estimates suggest that AFM is the dominant type of muscle fiber in insects, found in nearly 75% of flying species [30]. Additionally, the distribution of AFM among insect taxa suggests that asynchronous operation has evolved independently as many as 7–10 times [30,31]. The reason behind the surprising presence of AFM among insect species lies in its higher energy efficiency. The leading energy-demanding processes in contracting muscle fibers are posed by Ca^2+^ cycling and actomyosin ATPases, which are driven by Ca^2+^. As AFM usually contracts at high frequencies, with much lower Ca^2+^ cycling and sarcoplasmic reticulum investment [30,32], the energy demand per contraction cycle is much lower than in SFM. Therefore, AFM is more efficient and more powerful than SFM for high-frequency operation [22,23,33,34,35]. Despite the lower energy costs of Ca^2+^ cycling, the energy demand to sustain such high frequencies of AFM contractions is exceptionally high, with upper limits of mass-specific metabolic rates close to 1000 W/kg [36]. To accomplish this, AFM oxygen consumption rates usually rise 10–100 fold during rest to flight transition compared to ≈18-fold increase in mammalian muscles [37,38,39].

Most of the biological energy is made available in eukaryotes by the complete oxidation of nutrients within their mitochondria through OXPHOS. This process takes place essentially at the mitochondrial inner membrane, involving numerous protein complexes that perform sequential redox reactions, which generate a proton gradient (also known as the mitochondrial membrane potential or ΔΨ_m_) that is utilized for ATP production by F_1_F_o_-ATP synthase. A critical component of the electron transport system (ETS) is the cytochrome *c* oxidase (COX), which is the terminal enzyme complex of all aerobic metabolism and mediates the transfer of four electrons from reduced cytochrome *c* (cyt *c*) to molecular oxygen. Therefore, considering the key role of COX in energy homeostasis, the present work aims to briefly summarize the current understanding of how the structure and function of the COX complex associate and contribute to flying insect physiology. Comparative analyses of sequence and structural modeling of specific COX subunits in distinct insect species led us to hypothesize critical aspects that should be further explored in future studies and that might be relevant to better understand the biology of insect flight and dispersal.

### 1.2. Insect Flight Is Powered by Mitochondrial Metabolism

Although insects are the most diverse taxonomic group containing the largest number of species, studies dedicated to the understanding of insect’s energy metabolism remain largely unexplored (≈1.8% of all original papers on mitochondria found in PubMed by November 2020). Insects possess several unique metabolic features that go beyond using them (mostly *Drosophila*) as a model organism to understand human diseases [40]. For example, mitochondrial transport and oxidation of dicarboxylates, such as succinate, are quite limited in insects [41,42,43]. Additionally, succinate oxidation in adult insects is loosely coupled to the generation of ΔΨ_m_, resulting in low efficiency of mitochondrial ATP synthesis [44,45,46]. On the other hand, succinate oxidation is high in the midgut of tobacco hornworm larvae but severely reduced upon the commitment to the metamorphosis to pupae [47]. In addition, the metabolism of mitochondrial Ca^2+^ in insects strikingly differs from mammals as very low Ca^2+^ uptake and stimulation of respiratory rates are observed [45,48,49]. The study of insect metabolism was critical in identifying and defining the biological significance of the glycerol phosphate shuttle [14,26,50,51,52,53]. In this regard, insect mitochondria have a remarkably high capacity to oxidize G3P, which is only matched by mammalian brown adipose tissue [53,54]. Comparative analyses among distinct insect species revealed that G3P dehydrogenase (G3PDH) activity and G3P production are much higher than lactate dehydrogenase (LDH) activity and lactate levels during flight [50,51]. This strongly suggests that G3PDH activity plays a key role in re-oxidizing cytosolic NADH in insect flight muscles, which seems to allow intense contraction without acidification due to high lactate accumulation.

A key aspect is the substrate preference that insects use to support flight, which depends on several factors, including the type and duration of the flight activity, aging, and dietary strategies [16,18,19,20,21,55]. Regardless of the flight regimen, glucose, trehalose, diacylglycerol, and proline are the main substrates consumed to support insect’s flight [16,56,57,58,59,60,61,62,63,64,65,66,67]. While migratory insect species usually oxidize fatty acids in their long-range flights [67,68], hovering insects preferentially use proline and carbohydrates as main fuels [59,61]. A particular case is proline, which is used by different insect species not only as a fuel to support flight but also as a carbon source to stimulate pyruvate oxidation [59,60,61,62,63,64,65,66]. Indeed, proline acts as the main substrate to support flight activity in several insect species, including bumblebees and obligatory blood-feeding insect vectors of neglected tropical diseases [59,60,69]. Interestingly, the high rates of proline oxidation observed in blood-sucking insects seem to represent an important adaptation to blood-feeding habit associated with a protein-rich diet [59].

Insect flight muscles are remarkably unique in several aspects, from the morphological, physiological, and biochemical perspectives. For example, in *Drosophila* the structure of the tropomyosin binding protein troponin T (TnT) has a unique C-terminal extension of ≈70 residues rich in glutamic acid and is essentially expressed in flight and leg muscles [70]. Importantly, TnT is essential for *Drosophila* motor activity, as climbing and flight abilities were strongly affected by genetic removal of TnT, possibly by acting as a mechanism for Ca^2+^ buffering [70]. From the morphological perspective, insect flight muscles have remarkable architectural features suggesting that proximity of energy supplying (mitochondria) and demanding (myofibrils) sites would facilitate the flow of substrates to sustain such intense mechanical work during the flight. Structural hallmarks can also be extended at the organelle level, including the high mitochondrial abundance and density, which occupies ≈30–40% of cell volume [13,20,55,71]. The densely packed cristae, resulting from the striking degree of inner membrane juxtaposition, also emerges as a remarkable feature among insect flight muscles (Figure 1B) [20,23,71].

Hummingbird flight muscles have higher mitochondrial cristae densities as compared to insect flight muscle but have lower mitochondrial volume densities [71]. The metabolic consequence of such high tissue mitochondrial and cristae density is the extraordinary capacity of insect flight muscle to consume oxygen [21]. For example, in honeybees, respiratory rates relative to the mitochondrial volume are 2–3 times higher than observed in hummingbirds and 4–6 times higher than in mammals [71]. Thus, the enormous respiratory capacity of insect flight muscle is a consequence of their remarkable ability to produce ATP by means of OXPHOS.

### 1.3. Regulation of Mitochondrial Energy and Redox Metabolism in Flying Insects

To sustain the remarkable ATP turnover production rates, flying insects evolved two unique metabolic features: (*i*) many enzymes in flight muscle operate close to or at their V_max_ during maximal activity [72], and (*ii*) the cellular contents of some enzymes [*E*] are in larger excess compared to organisms with lower metabolic rates [73]. Indeed, reductions of ≈50% in the activity of several metabolic enzymes caused no apparent effects on *Drosophila* wing beat frequencies, supporting the concept of a large excess of enzyme content in insect flight muscles [73,74]. However, the fine-tuning of respiratory rates is critical for the regulation of mitochondrial oxidant production and flight dispersal in insects [8,20,21,75]. For example, in the hawkmoth *Manduca sexta*, flight ability and oxidative damage increase in flight muscle proteins upon sugar-feeding [75]. The females of the major arbovirus vector *Aedes aegypti* represent an extreme example of the metabolic and physiological adaptations to the diet in a flying insect. Adult *A. aegypti* females usually acquire their nutrients from plant sap, which is rich in sugar and some amino acids, including proline [76]. However, to trigger the gonotrophic cycle and egg production, females need to feed on vertebrate blood (that is why *A. aegypti* is classified as facultative hematophagous insects). Vertebrate blood is rich in proteins and lipids but poor in sugars and strikingly differs from the usual mosquito diet on plant sap. Therefore, this exquisite food digestion and metabolism represent a substantial biochemical challenge for hematophagous organisms [77]. To overcome this, the metabolism of specific amino acids (including proline and tyrosine) represents an important adaptation for the blood-feeding habit [59,69,78]. Mosquitoes have a high engorgement capacity, taking up close to three times their own weight on vertebrate blood per meal. It is thus remarkable their stupendous ability to fly even with such a high meal payload in their guts (imagine drinking a “smoothie” composed of ≈ 7 L water, ≈ 200 kg of meat, plus two tablespoons of sugar just before running a marathon!). The metabolic consequences of such a high protein, fat, and iron diet include the reduction of respiratory rates and mitochondrial oxidant production [20] and their flight activity [79]. The mitochondrial functional changes promoted by blood meal were a consequence of modulation of specific mechanisms, including the induction of mitochondrial fusion and specific reduction of COX activity [20]. These effects are fully reverted three days after a blood meal, indicating that blood-derived products act as signals to trigger mitochondrial functional and structural remodeling.

Energy metabolism is regulated by distinct factors and usually at multiple steps. In this regard, respiratory rates and mitochondrial ATP synthesis are classically regulated by three factors: (*i*) substrate supply, (*ii*) energy “wasting” by proton leak, and (*iii*) energy demand [80]. A classical model of metabolic regulation establishes that products of ATP hydrolysis, resulting from increased energy demand, bind to regulatory enzymes and activate catabolic pathways (the so-called “adenylate model of metabolic regulation”). Conversely, the flux of anabolic pathways is reduced by the same principle but on different enzyme targets. When energy demand ceases, decreased ATP utilization results in repression of catabolic pathways by ATP while activating anabolic pathways. The consequence would be a remarkable stability of cellular ATP levels (or energy homeostasis) even during massive energy demands such as during flight [81,82]. Mechanistically, two possibilities would explain how energy demand regulates respiratory rates and OXPHOS. The first one states that under high energy demand, ADP activates the F_1_F_o_ ATP synthase complex to produce ATP by using the energy from ΔΨ_m_ through a chemiosmotic mechanism [83]. As the ΔΨ_m_ magnitude directly affects the respiratory rates [83,84,85], the mechanism provided by the ATP/ADP ratio regulates respiratory rates in a ΔΨ_m_-dependent fashion [83,84,85]. The second possibility involves the allosteric regulation of several mitochondrial enzymes (including dehydrogenases and COX) by the ATP/ADP ratio independently of the ΔΨ_m_ [64,86,87,88,89,90]. Interestingly, both possibilities were experimentally demonstrated in flying insects, providing supporting evidence for the regulation of respiratory rates and OXPHOS in this particular group of organisms [64,86,90].

Despite its simplicity, the “adenylate model of metabolic regulation” does not seem to be the major driver for the maintenance of ATP turnover rates [73,91,92]. For example, it is long known that during insect flight, respiratory rates rise several hundred-fold but, paradoxically, adenylate levels hardly change [81,82]. Additionally, the desert locusts *Schistocerca gregaria* under anoxia show increased AMP, ADP, and Pi, but low ATP levels. This sharply contrasts with the postulate of the “adenylate model of metabolic regulation” since ATP turnover rates were not activated as it would be expected [91]. However, how would insect flight muscles increase respiratory rates without apparent changes in adenylate levels? A nice explanation is supported by experiments using *Drosophila* mutants to specific energy metabolism enzymes while assessing wing beat frequency as a proxy of flight capacity [73]. Reductions of up to ≈90% of most metabolic enzyme activities caused no apparent changes in wingbeat frequencies, strongly suggesting that insect flight muscle has an excess capacity of metabolic enzymes to sustain flight [73]. This observation fits with previous proposals suggesting that enzyme concentration [*E*] would be the strongest regulator of ATP turnover rates as seen in flying insects [72,92,93]. Indeed, hexokinase, phosphofructokinase, citrate synthase, and COX activities in honeybee flight muscle operate closer to V_max_ than in mammals [72]. Given that [*E*] is a key determinant for V_max_ and that activities of metabolic enzymes in insect flight muscle are much higher than in mammals [93], the remarkably high metabolic fluxes of insects during flight may be accomplished by having high [*E*] working closer to V_max_ than mammalian enzymes. Although metabolic fluxes seem to be mostly controlled by [*E*], this by no means excludes the regulatory roles of adenylates in fine-tuning other critical metabolic processes (such as the redox balance), as will be discussed later in this review.

## 2. Cytochrome *c* Oxidase in Flying Insects

### 2.1. A Brief Historical Background

Our understanding of COX is historically linked to insects since the seminal works of Charles MacMunn and David Keilin, who firstly identified histo/myohematins and later cytochromes using insects as models of study [94,95]. Although MacMunn has found “cytochromes” in testes and gut of different insect species, Keilin observed that honeybees’ thoracic muscles were “the best material for the study of the absorption spectrum of cytochrome” [94,95]. Keilin also noted that “among all organisms examined, the highest concentration of cytochrome is found in the thoracic muscles of flying insects”, indicating the feasibility of insect flight muscle for cytochrome studies [95]. This postulate was later substantiated in fruit flies and blowflies when the so-called “sarcosomes” were finally defined as mitochondria, given their strong similarities in structural and biochemical properties [96]. Indeed, the high cytochromes content in insect flight muscle indicates their role in biological oxidations and energy transduction and correlates with the enormous energy demand that flight activity poses to this unique tissue [96,97]. Key observations also revealed the association of cytochromes during metamorphosis and development, as their levels sharply and specifically rise in flight muscle upon pupal–adult transition [95,98,99]. By using the common wax moth *Galleria mellonella* attached to glass slides, Keilin also established cytochromes as entities involved in redox reactions of respiration and metabolism [95]. Over the years, the function of cytochromes and COX to insects revealed its multiple facets to physiology and biology [43,47,99,100,101].

### 2.2. Biological and Physiological Roles of COX to Flying Insects

Changes in several biological and physiological parameters, including shifts in dietary preferences, development of insecticide resistance, and aging were associated with altered COX function in flying insects [20,43,47,97,98,99,100,101,102,103,104,105,106,107,108,109,110,111,112,113,114]. For example, in *A. aegypti* mosquitoes, a change from sucrose to blood diet transiently reduced flight muscle cytochromes *a* + *a_3_* content and COX activity [20]. These events are parallel to reductions in respiratory rates and hydrogen peroxide production, which are all reverted right upon blood being fully digested. Conceivably, reductions in flight muscle mitochondrial metabolism triggered by blood meal would have two physiological consequences for mosquitoes: (*i*) to spare nutrients from flight muscle to ovaries as a mechanism to support oogenesis, and (*ii*) to avoid the potential oxidative burst generated by the interaction of blood-derived products such as heme and iron with hydrogen peroxide produced by mitochondrial metabolism [104,105]. The first possibility fits within the broad concept of “flight-oogenesis syndrome”, a physiological process by which some migrating insects alternate two energy-competing states: migration and reproduction [106], which is not the case for the mosquitoes. Alternatively, reductions in mitochondrial metabolism and oxidant production would represent a unique preventive antioxidant defense for mosquitoes (and other hematophagous organisms [20,104,105]) to avoid the potential toxicity of their unusual diet. In wood-fed *Anoplophora glabripennis* beetles, the expression of genes involved in protein metabolism and 13 subunits of COX was significantly higher than in artificial diet-fed insects [102]. This suggests that wood-feeding requires an increase of respiratory capacity in beetles, possibly to meet the energy demand posed by faster protein turnover. Associations between COX activity and insecticide resistance were also observed in flying insects. Strains of the housefly *Musca domestica* resistant to dichlorodiphenyltrichloroethane (DDT) exhibited higher COX activity compared to sensitive ones in a sex-independent manner [99]. In line with these observations, *Sitophilus oryzae* rice weevil and *Blattella germanica* cockroach strains resistant to insecticides have higher COX activity than nonexposed or susceptible individuals [107,108]. Regarding aging, specific reductions in COX function were observed in several flying insects along with their lifespan [43,109,110,111,112,113]. For example, selective downregulation of nuclear and mitochondrial COX subunits expression is associated with specific reductions in COX activity during *Drosophila* aging [43,111,112,113]. Interestingly, mitochondrial oxidant production is promoted by inhibition of COX activity by using specific drugs in *Drosophila* and *Musca* [43,110]. On the other hand, *Drosophila* COX activity is reduced by redox imbalance, a pattern that is linked to specific degeneration of mitochondrial cristae [113,114]. Concerning the development, flight muscle mitochondria undergo important morphological changes, including the development of densely packed lamellar cristae as well as an increase in COX activity upon *Drosophila* emergence to adult stages [115]. Interestingly, COX activity and COXIV expression were essentially found at lamellar cristae, strengthening the concept that cristae are the true bioenergetic units responsible for mitochondrial energy transduction [116].

### 2.3. Regulation of COX Activity in Flying Insects

Evidence suggests that flight metabolism and dispersal potential are tightly linked to COX function. For example, long-distance migratory butterfly species have higher COX content and activity than short-distance fliers [117]. In this regard, the migratory butterfly *Vanessa atalanta* flight muscle mitochondrial area and cristae density were higher compared to the short-range butterfly *Melitaea cinxia* [117]. Remarkably, the relationship between dispersal potential and COX activity can also be observed within the same flying insect species. Recently established populations of *M. cinxia* butterflies have higher dispersal potential than old ones, a phenotype that is mirrored in COX activity [117]. This strongly indicates that COX represents a key metabolic mechanism for dispersal potential in flying insects.

Regarding the factors that regulate COX activity in flying insects, scarce information is available. Some of the known regulators of COX include oxygen availability, hormonal signaling, redox homeostasis, and adenylate balance [90,118,119,120,121,122]. For example, Tibetan highlander populations of migratory locusts, which are naturally exposed to low apO_2_, preserve mitochondrial integrity, ΔΨ_m_, COX activity, and turnover upon hypoxia exposure [118]. On the other hand, lowlander populations exhibit considerable changes in mitochondrial functionality upon hypoxia [118]. Remarkably, highlander populations also exhibited altered kinetics of COX, including higher affinity and V_max_ than lowlanders. This indicates that adaptations of flying insects to low apO_2_ involve the regulation of COX by increasing its catalytic efficiency rather than its content [118]. Metalation of COX Cu_A_ sites is mediated by “syntheses of cytochrome *c* oxidase” (SCO) protein, which is absolutely required for COX assembly and function. In the eastern honeybee *Apis cerana*, COX activity requires SCO, whose expression is induced by different stress signals, including cold, transition metals, ultraviolet, and oxidant exposures [120].

Since COX activity is regulated by the adenylate balance in vertebrates [88,89], an emerging question would be whether the flux of catabolic pathways can be controlled by allosteric regulation of COX activity in flying insects? Indeed, a direct demonstration of allosteric regulation of COX by adenylates in invertebrates was missing. Recently, our laboratory described for the first time that in insects, COX is allosterically regulated by adenylate balance, with direct consequences on mitochondrial metabolism [90]. We observed that ADP activates and ATP strongly inhibits respiratory rates sustained by G3P oxidation in *A. aegypti* flight muscle [90]. These effects were independent of substrate transport to mitochondria, TCA cycle enzyme activities, and ΔΨ_m_. In addition, we observed that regulatory effects of adenylates were specific to COX activity and not to other ETS components. Indeed, ATP not only exerted powerful inhibitory effects on *A. aegypti* COX, reducing its activity by ≈75% but also shifted the enzyme kinetics from a typical Michaelian hyperbolic-shaped curve to a sigmoidal one. This suggests that at least in *A. aegypti* mosquitoes, ATP promotes conformational changes in COX structure, possibly favoring the interaction of two complex monomers and resulting in positive cooperativity between the cyt *c* binding sites. Finally, the regulatory sites of adenylates on respiration seem to involve targets facing both the intermembrane space and matrix, as observed in vertebrates [88,90,121,122]. To our knowledge, this was the first evidence demonstrating allosteric regulation of COX by adenylates in invertebrates, underscoring this mechanism as a much broader and evolutionary conserved process of energy metabolism regulation than previously thought.

## 3. Phosphorylation as a Critical Mechanism of Post-Translational Modification and Regulation of COX

COX is a multimeric protein complex composed of three subunits encoded by the mitochondrial DNA (representing the catalytic enzyme core) and 11 subunits encoded by the nuclear genome in eukaryotes [123,124]. Given that COX represents a key site of OXPHOS regulation [125], it is not surprising that in vertebrates it is under the control of several regulatory mechanisms at transcriptional, post-translational, and allosteric levels [88,89,90,121,122,126,127,128,129,130,131,132,133,134,135,136,137,138,139,140,141,142,143,144,145].

Phosphorylation of specific Ser/Thr and Tyr residues in different COX subunits represents the best known post-translational modifications of this enzyme complex. Notably, several lines of evidence support the notion that phosphorylation of specific subunits directly modulates COX function in vertebrates, essentially by sensitizing this enzyme complex to the allosteric effect of adenylates [133,134,135,136,137,138,139,140,141,142,143,144,145]. The first observation of COX phosphorylation identified a 17k Da protein as the major COX phosphoprotein in rat heart and liver mitochondria [133]. Subsequently, this 17k Da phosphoprotein was identified as the subunit IV of COX (COXIV) [133]. Mechanistically, cAMP-dependent protein kinase (PKA) phosphorylation of specific residues in different COX subunits renders the enzyme complex more prone to allosteric ATP inhibition, which is antagonized by dephosphorylation mediated by protein phosphatases [135,136,140]. PKA-dependent phosphorylation of COXII and COXV were also reported in bovine heart mitochondria, a step necessary for allosteric ATP-inhibition of COX activity [136]. Strengthening these observations, the presence of protein phosphatase 1 (or its activator, Ca^2+^) reverted the inhibitory effects of PKA-dependent phosphorylation of COX [136]. In cow liver and heart, PKA-dependent phosphorylation of Tyr residues of subunit I change the COX kinetics in a way that renders the enzyme sensitive to ATP inhibition [141]. In murine macrophages and rabbit hearts, hypoxic challenges induce PKA-dependent phosphorylation of subunits I, IV-1, and Vb [142,143]. Importantly, mitochondrial oxidants activate PKA during hypoxia [144] and phosphorylation of COX subunits reduced its activity, which in turn increased oxidant production [142].

COX phosphorylation can also be mediated by PKCε, which associates with subunit IV in neonatal cardiac cells upon hypoxic preconditioning [137,138,139]. However, PKCε-dependent phosphorylation increases COX activity, which contrasts with the functional outcome of PKA-mediated phosphorylation of COX [135,136,137,138,139,145]. This indicates that phosphorylation of specific residues confers unique functional outcomes for COX activity. COX is also regulated by proinflammatory cytokines such as tumor necrosis factor-alpha, which mediates Tyr phosphorylation of COXI, reducing its activity and compromising ΔΨ_m_ and OXPHOS [145].

An interesting hypothesis suggests that allosteric regulation of COX activity acts as a mechanism to control ∆Ψ_m_ and reactive oxygen species (ROS) production. This postulate states that COX phosphorylation renders the enzyme sensitive to allosteric ATP inhibition and maintains ∆Ψ_m_ in levels lower enough to preserve ATP synthesis (100–140 mV), but drastically affecting mitochondrial oxidant production [140,146]. Indeed, increased mitochondrial ATP/ADP ratios reduced ∆Ψ_m_ to ≈ 133 mV in the liver, but not in heart mitochondria [146]. Although attractive, further research is needed to unequivocally demonstrate the consequences of COX phosphorylation and regulation by adenylates on mitochondrial superoxide production, particularly during work–rest transitions.

Although numerous phosphorylation sites on different COX subunits have been predicted over the years, only some of them were validated while new sites were found. For example, in a comprehensive tissue catalog phosphoproteome of 14 rat tissues identified eight phosphorylation sites in COXIV, two in COXVIc, and one in COXII [147]. A remarkable consequence of COX phosphorylation is the transition from the monomeric to the dimeric state of the enzyme complex [127]. Conversely, high cytosolic Ca^2+^ activates protein phosphatases which dephosphorylate COX and shifts the enzyme complex from the dimeric to the monomeric state [127,136,140]. The metabolic consequences of this transition include the blockage of the allosteric ATP inhibition of COX, reduced efficiency of electron transfer, increased ΔΨ_m_ and mitochondrial oxidant production [127,136,140]. Since most COX structures solved by X-ray crystallography are in the dimeric form, this strongly indicates that reversible phosphorylation has a crucial role in determining the enzyme kinetics and activity. COX dimerization mediated by reversible phosphorylation might also explain the shift from hyperbolic Michaelian to sigmoidal allosteric kinetics induced by high ATP/ADP ratios and PKA-dependent phosphorylation, as cooperativity intrinsically requires at least an enzyme dimer to allow this regulation [127]. A beautiful example of regulation of COX activity identified a signaling cascade in which the TCA-derived carbon dioxide (CO_2_) activates soluble adenylate cyclase (sAC) and the PKA-dependent phosphorylation of COXI and COXIV-2 culminating with improved COX activity [148]. The consequences of COX phosphorylation through this signaling cascade are the activation of OXPHOS and reduction of mitochondrial oxidant production [148]. Likewise, the metabolic effects of the CO_2_-sAC-cAMP-PKA signaling pathway were also observed in yeast, directly regulating COX activity and OXPHOS and underscoring the evolutionary conservation of this mechanism for metabolic regulation [149]. A further development identified COXIV-1 as a key target of PKA-dependent phosphorylation and regulation of COX activity by adenylates [150]. Specifically, Ser58 was the dominant site of COXIV-1 phosphorylation by PKA, and the exchange of this residue to a phosphomimetic aspartate increased COX activity while rendered the enzyme insensitive to adenylate regulation [150]. This observation contrasts with the experiments carried out in yeast *COX5a*, the mammalian orthologue of COXIV-1, where the substitution of Ser51 to Asp caused no apparent effects on COX activity and adenylate regulation [149]. However, changing Ser43 or Thr65 to Asp decreased COX activation by ADP while did not alter ATP inhibition of COX [149]. Therefore, maintaining a negatively charged environment provided by a phosphomimetic amino acid at critical phosphorylation sites seems to confer reduced COX sensitivity to allosteric regulation by adenylates, whether in yeasts or mammals [149,150].

It is long known that allosteric regulation of COX activity by adenylates plays a key role in controlling the electron flow through the ETS complexes [88,89,121,122,123,132]. Several lines of evidence demonstrate that adenylates act as specific and critical regulators of COX function in yeasts and vertebrates. Although most prokaryotes only code the core catalytic subunits of COX (subunits I-III), rendering bacterial COX activity insensitive to changes in ATP/ADP ratios [151], some prokaryote genomes code for additional COX subunits. For example, the cyanobacteria *Synechocystis* sp. codes for an additional proto-COXIV subunit that allowed adenylate regulation of COX activity [152]. The first observations on adenylate regulation demonstrated that in several vertebrate species, ATP binds to COX, reducing its affinity to cyt *c*, and ultimately decreasing its activity [153,154].

Interestingly, the specificity of nucleotide regulation of COX seems to be strong for adenylates as neither cytidine, guanosine, nor uridine-nucleotides promote conformational changes in the enzyme complex [121]. Subsequent research identified that adenylates regulate COX activity by binding to specific sites at subunits IV and VI [88,89,121,122,123,132,150,155,156,157,158]. While the association of ATP/ADP to subunit IV takes place at both sides of the mitochondrial inner membrane [89,122,150], subunit VI binds adenylate only at the matrix side [155,157]. The body of evidence accumulated clearly indicates that energy demand, through changes in ATP/ADP ratio, allosterically regulates COX activity and impacts respiratory rates.

To our knowledge, the only available evidence that adenylates regulate COX activity in insects was reported for the major arbovirus vector *A. aegypti* mosquitoes [90]. We observed that ATP/ADP controls mitochondrial G3P oxidation through specific and allosteric regulation of COX activity. Additionally, inhibition of the adenine nucleotide translocator decreases by ≈ 50% the activating effects of ADP on uncoupled respiratory rates without affecting their apparent affinity. This strongly suggests the existence of two distinct sites by which adenylates regulate respiration: one at the matrix and the other at the intermembrane space [90]. Although the exact sites of adenylate binding in flying insects are yet to be determined, we postulate that subunits IV and VI are the most likely candidates to regulate COX activity.

To further investigate this possibility, we next provide comparative bioinformatic analyses of three COX subunits from several insect and mammalian species. Since our laboratory recently demonstrated that COX activity in a flying insect is allosterically regulated by adenylates [90], we hypothesized that COXIV and COXVI in these organisms would have unique features as potentially relevant mechanisms to regulate energy and redox metabolism. This would be particularly interesting considering the existence of allosteric regulation of COX activity by adenylates in metabolically extreme models such as in flying insects [90]. We then focused our analyses on COXIV and VI because adenylate binding and the regulation of COX activity are historically and consistently described for these subunits more than any other COX component [159].

### 3.1. Structural Features of Insect COX Subunit IV

COXIV is the largest subunit of the COX complex encoded by the nuclear genome and contains two domains facing the mitochondrial matrix and the intermembrane space, as well as a transmembrane helix domain that crosses the mitochondrial inner membrane [160]. The C-terminal of COXIV interacts with COXII and contributes to cyt *c* binding, while the transmembrane domain tightly associates with COXI [160]. Moreover, the C-terminal domain is one of the adenylate binding sites that mediate the allosteric regulation of COX activity [122]. COXIV subunit is essential to the assembly of COX complex, which is initiated by the association of COXI and COXIV. Indeed, silencing COXIV significantly reduced COX content, respiration, and cell viability [161,162]. The N-terminal domain of COXIV faces toward the mitochondrial matrix and acts as the second site of allosteric regulation by adenylates [89]. In mammals and other organisms, COXIV is found as two distinct isoforms (COXIV-1 and COXIV-2), which diverged ≈320 Mya, by the time the highest apO_2_ and the presence of giant flying insects in the late Paleozoic took place. In this regard, despite the COXIV-1 and -2 high similarity, key differences emerge at the matrix-facing N-terminal region. First, the N-terminal of COXIV-2 lacks the regulatory Ser 58 phosphorylation site, which may reduce its regulation by adenylates [150]. Second, three cysteine residues in the COXIV-2 N-terminal region may confer redox regulation of COX activity [163]. Interestingly, two of these cysteine residues are close to the proposed adenylate binding site [164]. Tissue distribution of COXIV isoforms in vertebrates revealed that COXIV-2 is highly expressed in the lung, while COXIV-1 has a ubiquitous distribution [163]. Importantly, loss of COXIV-2 impairs lung physiology [164], underscoring that COXIV isoform expression may serve as a mechanism for oxygen sensing and adaptation to hypoxic challenges [159,165]. Supporting evidence suggests that the divergence of two COXIV isoforms has parallels with their distinct functions. For example, the expression of COXIV isoforms is regulated by apO_2_ in a way that COXIV-2 expression is activated during hypoxia through a conserved hypoxia-sensitive element [165]. This agrees with kinetic analyses of COXIV-2 showing lower affinities for oxygen as compared to COXIV-1 [166]. Regarding the adenylate regulation, molecular modeling on mammalian COXIV-1 suggested that γ/β-phosphates of ATP would interact with the N-terminal Ser56 and Ser58 side chains located at the matrix side [150]. As pointed out earlier, Ser58 acts as a key site of PKA-mediated phosphorylation site that prevents the inhibitory effect of ATP on COX activity [150].

To explore the potential biological roles of COX in insects, we searched COXIV sequences from 24 insect species with sequenced genomes and compared them with those from all mammalian species included in the Uniprot database. The protein prediction of the insect genomes was searched using the COXIV conserved domain (PF02936) and the HMMER3 software [167]. We also searched the mitochondrial targeting signal (MTS) by using the TargetP 2.0 package [168]. Sequences that lacked an MTS were longer than 50% average sequences, and with nonaligned segments were removed from our analyses. The Uniprot mammalian proteins were searched using web NCBI blastp with the Uniprot database (filter “mammalian”) and *Mus musculus* COXIV-1 (P19783) and COXIV-2 (Q91W29). All the COXIV protein sequences identified were aligned using the web Clustal Omega tool [169]. We observed that COXIV sequences from insects retrieved from our analyses had very high degrees of homologies with mammalian COXIV. However, we could not define the specific COXIV isoforms (1 or 2) for any insect species. The presence of a variable number of COXIV isoforms, together with the identification of protein sequences with variable length (usually caused by automated gene prediction mistakes), lead to a scenario of uncertainty about the isoform numbering. Notwithstanding, key features of insect COXIV sequences were observed. For example, we identified an exchange of the Ser58 conserved in mammals to a phosphomimetic Glu or Asp residue in the N-terminal region of COXIV in most insect species analyzed (Figure 2B). The presence of a Glu or Asp in the N-terminal region of insects COXIV (instead of a canonical Ser in mammals [148]) suggests that regulation of COX activity by adenylates through this site is not evolutionarily conserved. Indeed, the Ser58 residue is not present in COXIV sequences from birds, fishes, and amphibians [148]. Considering that in mammals, this Ser at the N-terminal region is a phosphorylation target that turns off allosteric ATP inhibition of COX [148] and that in most insect species this residue is changed to a phosphomimetic Glu or Asp, two possible explanations emerge to explain the allosteric inhibitory effect of ATP on insect COX activity [90]: (*i*) that it does not involve the canonical N-terminal region of COXIV [150] and/or (*ii*) that regulation of COX activity by adenylates is mediated by a distinct (and still elusive) site at COXIV. To address these possibilities, we generated tridimensional structure models of COXIV from eight insect species by using the Phyre2 web portal [170] and the bovine COXIV (1V54) as a template [171]. We observed that all eight sequences of insect COXIV nicely overlapped with bovine COXIV, suggesting that differences in amino acid sequences between insects and mammals are, in general, not reflected on overall COXIV structure (Figure 2A). Interestingly, we observed a unique extra β-strand at the N-terminal region of the honeybee *Apis mellifera* COXIV structure model. Noteworthy is the strong overlap of all insect COXIV structural models at both matrix N-terminal and intermembrane space C-terminal regions, which are the most likely sites of phosphorylation and regulation by adenylate binding. This suggests that if regulation of COX activity through adenylate binding at COXIV is mediated by an elusive site, this would probably involve only the C-terminal region facing the intermembrane space. If that is the case, this would be particularly critical for ATP-mediated inhibition of COX activity as phosphomimetics prevent this effect in mammalian COX [150]. However, we previously reported that ATP exerted powerful inhibitory effects on both COX activity and respiration in *A. aegypti* flight muscle mitochondria [90]. Importantly, our results suggest the activating effects of ADP on uncoupled respiratory rates in *A. aegypti* involve sites located at both mitochondrial matrix and the intermembrane space compartments [90].

We next predicted the potential phosphorylation for 38 COXIV sequences from 24 insect species by using the NetPhos 3.1 server. For these analyses, the phospho-sites (P-sites) pSer30, pTyr33, pSer56, pSer58, pSer69, pSer72, and pSer74 described for *Bos taurus* [172] were searched in insect sequences. We observed that pSer30 and pSer72 sites present in mammalian COXIV were changed to amino acids that cannot be phosphorylated in ≈95% of COXIV insect sequences analyzed. Indeed, two out of the 38 insect COXIV sequences have a pTyr30 instead of a pSer30, while all other COXIV sequences had neither a non-phosphorylatable nor a phosphomimetic amino acid at this site. Noteworthy, the vast majority of insect COXIV sequences analyzed do not have a predicted site for phosphorylation at pSer58 and pSer69 as these amino acids are substituted by a phosphomimetic Glu or Asp residue (Figure 2B).

Insects exhibit distinct flight regimens, including wingless and short-range species and migratory species that cover large distances. In this regard, we observed that the Tyr33 site was differently distributed among insects depending on their flight regimen. For example, the pTyr33 site was present in six out of seven COXIV sequences from the flightless species analyzed. On the other hand, this P-site was absent in 75% of short-range flying species, including mosquitoes, fruit flies, and houseflies. Similarly, the pTyr site is also absent in 80% of COXIV sequences from hovering species, including honeybees and bumblebees. On the other hand, migratory species, including the monarch butterfly *Danaus plexippus* have at least one COXIV sequence with the predicted pTyr. Therefore, given that this was the only P-site with differences among the different fly patterns, we hypothesize that pTyr33 might play a role in regulating COX activity for optimal fly behavior.

In general, the results of phosphorylation prediction and the presence of phosphomimetic amino acids suggest more similarities of the insect COXIV with mammalian COXIV-2, which includes the absence of the pSer30 and pSer72 sites and the presence of a phosphomimetic Glu or Asp at position 58. Nevertheless, the presence of Glu or Asp at position 69 on insect COXIV is a feature of mammalian COXIV-1. Conceivably, insects have different mechanisms to avoid oxygen toxicity, including not only the cyclic opening of tracheal spiracles [8] but also the expression of a COXIV subunit with a lower affinity for oxygen as observed in mammalian lungs [165,166]. However, it is important to remind that functional consequences of COX phosphorylation are site and/or subunit-specific. While PKA-mediated phosphorylation of COXIV-1 prevents ATP inhibition of COX activity [150], phosphorylation of other subunits causes an opposite effect [136]. Curiously, the wingless insects showed all of their COXIV with predicted phosphorylation at this residue. In addition, it is possible that the presence of the two phosphomimetic positions (58 and 69) might impair the binding and the inhibitory effect of ATP on COX activity. Again, the functional consequences provided by the presence of two phosphomimetics instead of a phosphorylatable Ser at the N-terminal region would maintain COX active regardless of the ATP/ADP levels. Conceivably, this would represent a mechanism developed by insects to avoid overwhelming mitochondrial oxidant production in conditions of low energy demand (such as during rest). Alternatively, avoiding COX inhibition by high ATP at this site might allow the energy to be dissipated as heat (thermogenesis) instead of being conserved as ATP. This would be particularly relevant in extreme cold conditions to allow insect dispersal and survival [26].

### 3.2. Structural Features of Insect COX Subunit VI

The COXVI subunits are also encoded by the nuclear genome and in most eukaryotes are found as three distinct genes, *COXVIa*, *COXVIb*, and *COXVIc* [159]. COXVIa and COXVIc proteins are evolutionarily related and transmembrane subunits [155,173], whereas COXVIb is the only subunit facing the intermembrane space that lacks a transmembrane helix [160]. Importantly, cyt *c* binding and COX dimerization seem to be mediated by COXVIb [160,174]. While scarce information is available for COXVIc [159,172], strong evidence indicates that COXVIa has two functionally distinct isoforms: the muscle (COXVIa-H) and the liver (COXVIa-L) [121,128,155,156,175,176,177,178]. Noteworthy, COXVIa is the target of different COX modulators, including palmitate and adenylates, with direct consequences to respiration [121,128,155]. Indeed, this ADP binding site at COXVI was confirmed by modeling a structurally related molecule (cholate) on bovine COX crystal [160]. Remarkably, a phosphorylated Thr residue (pThr23) located at the beginning of the transmembrane α-helix facing the N-terminal matrix side of the bovine COXVIa was observed in the crystal structure [171]. The stimulation of COX activity by ADP depends in part on its binding to the N-terminal matrix side of COXVIa [155]. Interestingly, this effect was specific for COXVIa-H as the activity of the liver isoform was not altered by ADP [155]. In addition, decreases on COX H^+^/e^−^ stoichiometry were observed under high ATP/ADP ratios, which was attributed to ATP binding to the N-terminal side of COXVIa-H [156]. Considering the energy conservation, low COX H^+^/e^−^ stoichiometries induced by high ATP/ADP ratios were proposed as a new thermogenic mechanism, particularly in conditions of reduced energy demand [176,177,178]. If that is the case, the thermogenic role of COX induced by high ATP would be specific for COXVIa-H as the H^+^/e^−^ stoichiometry of COXVIa-L is intrinsically low and insensitive to adenylate regulation [177,178]. Conceivably, reduced COX energy efficiency provided by adenylates might also have implications on redox homeostasis by affecting mitochondrial oxidant production.

In order to gain further insights into the potential biological roles of COXVIa to insects, we searched COXVIa sequences from 23 insects and performed comparative analyses with bovine COXVIa as described earlier for COXIV. We observed that COXVIa sequences from insects retrieved from our analyses had very high homologies with mammalian orthologs. Additionally, we could not define the specific COXVIa isoforms (H or L) for any insect species. We generated tridimensional structure models of COXVIa from nine insect species by using the Phyre2 web portal [170] and the bovine COXVIa-H (1V54) template as described earlier [171] (Figure 3A). We observed that all insect structure models of COXVIa nicely overlapped with bovine COXVIa, suggesting that slight differences in amino acid sequences between insects and mammals are in general not reflected on COXVIa structure (Figure 3A). Interestingly, we observed that a conserved P-site (Ser/Thr/Tyr) located at the beginning of the transmembrane α-helix facing the N-terminal matrix side was found in 17 out of 23 insect species analyzed, as well as in the bovine COXVIa (box in Figure 3A,B). Considering that allosteric regulation of COX by adenylates involves their binding to the N-terminal matrix side of COXVIa [155,157,159], which is modulated by phosphorylation of specific residues in different COX subunits [133,134,135,136,137,138,139,140,141,142,143,144,145], our analyses led us to hypothesize that the evolutionarily conserved P-site at the beginning of the transmembrane α-helix of COXVIa might play a role in regulating COX activity in both insects and mammals.

The subunit VIc of the COX complex is a transmembrane protein with a small α-helix domain facing the intermembrane space that closely interacts with COXII [159,160]. On the matrix side, COXVIc has a disordered region that does not seem to interact with any other COX subunit. Although no specific functions were attributed to COXVIc, two phosphorylation sites in COXVIc were identified by mass spectrometry in the rat brain, heart, kidney, and intestine [147].

We searched COXVIc sequences from 19 insect species with sequenced genomes and compared them with bovine COXVIc. For these comparisons, we essentially used the same approach as described earlier for COXIV and COXVIa. We observed that the predicted insect COXVIc tertiary structure models are very similar to mammalian COXVIc orthologs (Figure 4A), although the identity of the sequences in insects is ≈40% comparing with the bovine. Additionally, the identified phosphorylated Ser at the N-terminal (Figure 4C,D, position 2) [147] is not conserved in about 2/3 of the studied insect species. Notwithstanding, almost all of these 2/3 species have a Thr with predicted phosphorylation and/or a phosphomimetic Asp/Glu between positions 2 and 5 (Figure 4D). Curiously, the COXVIc sequences of *C. felis*, a wingless species, do not have the N-terminal Ser, Thr, or an acidic amino acid. The other wingless species, the silkworm moth *B. mori*, has only a phosphomimetic Glu at position 5. This raises the possibility that in insects, the absence of this phosphorylation site (or the presence of a phosphomimetic amino acid) might keep COX in a chronically activated state, avoiding the allosteric regulation of adenylates. This might also indicate that insect species that code COXVIc with a phosphomimetic amino acid at the N-terminal are more susceptible to mitochondrial oxidant production, as the allosteric adenylate control can be absent. A conserved phosphorylation site at Ser is also present in the C-terminal of both mammal and nonmigratory insects (Figure 4B, top and middle). The predicted phosphorylated Ser residue at the C-terminal in insects corroborates the identification of this post-translational modification by mass spectrometry [147]. Remarkably, this Ser residue is changed to either Thr or to a phosphomimetic Asp only in the migratory insects *Locusta migratoria* and *Nilaparvata lugens*, respectively. On the other hand, the Ser at the same position in the C-terminus is not conserved in migratory insects, which can be changed to a potentially phosphorylated Thr or a phosphomimetic Asp residues (Figure 4B, bottom). We hypothesize that the adenylate regulation of COX activity might be impaired in migratory insects that code COXIVc with a phosphomimetic residue at the C-terminal region, thus leaving the enzyme complex active and with potential consequences for redox homeostasis.

## 4. Conclusions and Future Perspectives

Beyond the classical role on energy metabolism, mitochondria are also critical regulators of cellular redox balance, acting as a key site of oxidant production at multiple sites [179]. As molecular oxygen is fully reduced to water by COX activity, electron leakage and oxidant production are unlikely to take place within the enzyme complex [180]. In this regard, the allosteric regulation of COX activity provided by adenylates emerged as a novel mechanism to control not only the energy metabolism but also the redox balance [88,89,90]. In this regard, the allosteric inhibition of COX by ATP was proposed as a mechanism to keep ΔΨ_m_ in levels lower enough to avoid mitochondrial oxidant production while preserving ATP production [128,140,146]. However, studies to substantiate this proposal are still scarce and the emerging picture about the involvement of COX on redox homeostasis is controversial. A study performed in HeLa cells demonstrated that PKA-dependent phosphorylation of COXI and COXIV-2 increased COX activity resulting in the activation of OXPHOS and reduction of mitochondrial oxidant production [148]. On the other hand, a study performed in rat heart apparently contrasts with these observations as PKA-mediated phosphorylation of COX was shown to mediate mitochondrial oxidant production [142]. Therefore, further research is required to directly demonstrate the contribution of allosteric regulation of COX on mitochondrial redox balance.

Flight has a critical role in allowing insect survival in almost all known ecological niches. To accomplish the huge energy demands associated with dispersal, flying insects are absolutely dependent on mitochondrial metabolism and OXPHOS. Since COX activity in insects is controlled by several factors, including nutrient supply and energy demand [20,90], one might speculate about the potential biological consequences of this regulation to flying insects. Conceivably, the allosteric regulation of COX activity by adenylates might act as a fine-tuning mechanism to modulate mitochondrial oxidant production in insects. In this sense, the consequences of altered COX function to mitochondrial redox balance were experimentally demonstrated in several insect species [20,43,110]. For example, while inhibition of COX activity boosts mitochondrial oxidant production in *Drosophila* and *Musca* [43,110], blood-feeding reduced COX activity and mitochondrial superoxide production in *Aedes* mosquitoes [20]. Importantly, respiratory rates and COX activity are strongly modulated by adenylates in the *Aedes* flight muscle, representing the very first example of adenylate regulation of COX in invertebrates [90]. Collectively, the literature search and the bioinformatic analyses conducted here suggest that insect COXIV and COXVI sequences and structure models are in general very similar to their mammalian counterparts. However, specific substitutions and exchanges to phosphomimetic amino acids at critical P-sites are hallmarks identified in both COXIV and COXVI insect sequences. Therefore, we envisage that the role of critical phosphomimetic amino acid residues at COXIV and COXVI should be further explored as potentially novel mechanisms to better understand the biological consequences of COX regulation in models of extreme metabolism.

## Figures and Tables

**Figure 1 cells-10-00470-f001:**
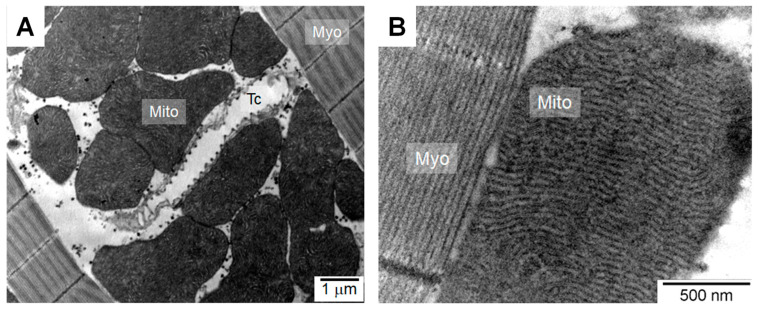
Transmission electron microscopy images of asynchronous flight muscles (AFM) biopsies from *Aedes aegypti* female mosquitoes. (**A**) The typical architecture of AFM longitudinal sections showing the high mitochondria (Mito) density localized alongside the myofibrils (Myo). Oxygen delivery to myocytes is made through a complex tracheae (Tc) network that lies adjacent to mitochondria. (**B**) Noteworthy is the extremely high cristae density and apposition, with barely visible matrix space.

**Figure 2 cells-10-00470-f002:**
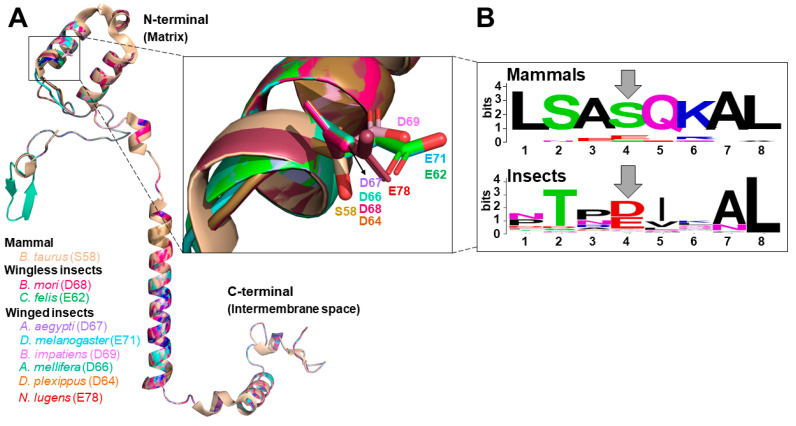
The tridimensional structure models of insects COXIV. (**A**) Overlay of eight structures of COXIV from insects modeled using the Phyre 2 server, using the primary sequence as input, normal modeling mode and the *Bos taurus* crystal structure (PDB: 1V54) as a template [171]. All structures were aligned from the mitochondrial matrix N-terminal on the top to the C-terminal at the intermembrane space at the bottom. The structures were analyzed using Pymol software and colored as shown in the figure. The conserved Asp (D) and Glu (E) residues in the α1-helix of insects COXIV are highlighted and shown in the inset box. Note that the only Ser residue (Ser58) identified is from *B. taurus*. (**B**) WebLogo diagram of α1-helix region close to mammalian Ser58 identified by the grey arrow. WebLogos of other insect COXIV regions containing known P-sites [172] are identified by the grey arrow. The WebLogos used the multiple alignment from 22 sequences from 19 mammals and 38 sequences of 21 insect species. Note the conserved Ser residue in the position 4 for mammals, equivalent to Ser58 [150], while for most insects this residue is exchanged for a Glu or Asp at the same position.

**Figure 3 cells-10-00470-f003:**
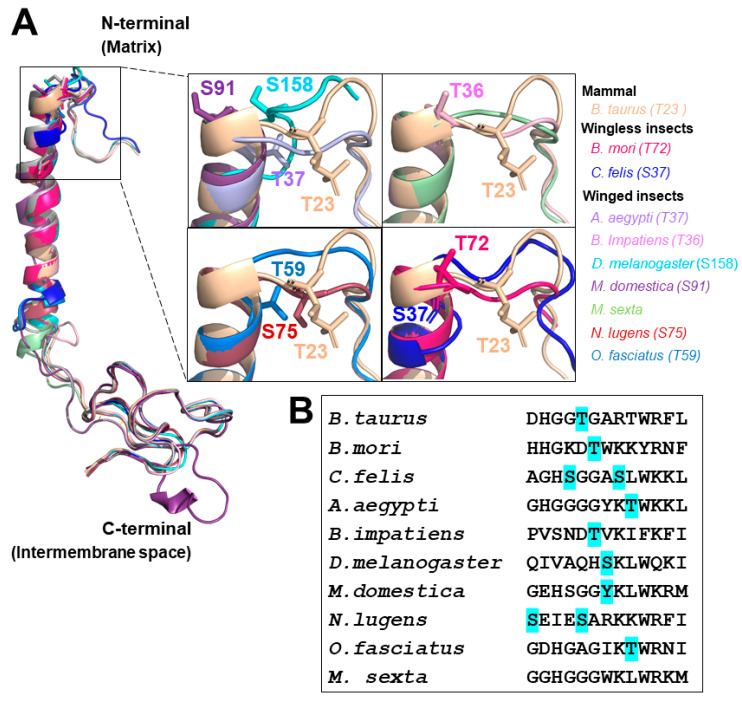
The tridimensional structure models of insect COXVIa: (**A**) overlay of nine structure models of insects COXVIa modeled in the Phyre 2 server using the primary sequence as input, normal modeling mode and the *B. taurus* crystal structures (PDB 1V54 or 3WG7) as templates [171]. All structures were aligned from the mitochondrial matrix N-terminal on the top to the C-terminal at the intermembrane space at the bottom. The structure models were analyzed using Pymol software and colored as shown in the figure. The positional conservation of the Ser and Thr residues at the beginning of a transmembrane helix is highlighted and shown in the inset boxes. (**B**) Primary sequence alignment fragment representing the beginning of COXVIa transmembrane helix. The highlighted Ser, Thr, and Tyr residues represent the NetPhos predicted P-sites.

**Figure 4 cells-10-00470-f004:**
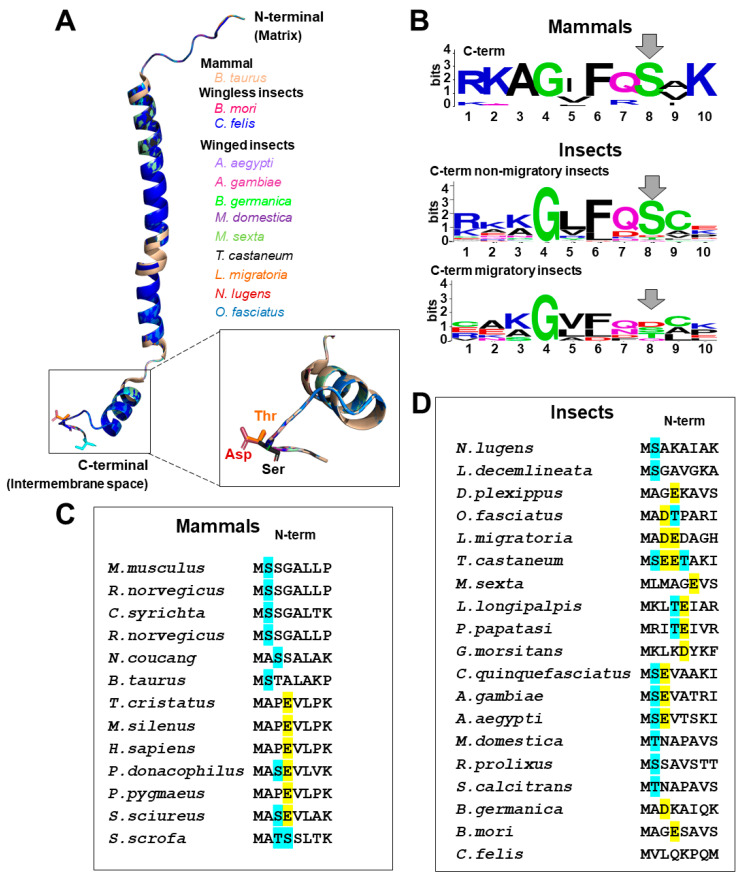
The tridimensional structure models of COXVIc in insects: (**A**) overlay of structure models of COXVIc from 11 insect species modeled in the Phyre 2 server using the primary sequence as input, normal modeling mode and the *B. taurus* crystal structure (PDB: 1V54) as a template [171]. All structures were aligned from the mitochondrial matrix N-terminal on the top to the C-terminal at the intermembrane space at the bottom. The structure models were analyzed using Pymol software and colored as shown in the figure. The conserved Ser residue at C-terminal is highlighted and shown in the box. Remarkably, the migratory insect species (*L. migratoria* and *N. lugens*) have a Thr or Asp instead of Ser residue, respectively. (**B**) WebLogo of a C-terminal region of mammals (top panel), nonmigratory insects (middle panel), and migratory insects (bottom panel). The WebLogo position 8 shows the conserved Ser for mammals (position 71 at *B. taurus* 1V54) and nonmigratory insects, while migratory insects also have Thr and Asp residues. The N-terminal of mammalian (**C**) and insects (**D**) COXVIc demonstrate the abundance of Ser and Thr with predicted phosphorylation sites (cyan) together with the phosphomimetic Asp or Glu (yellow).

## Data Availability

Not applicable.

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
