# Peer review of "Cytochrome *c* Oxidase at Full Thrust: Regulation and Biological Consequences to Flying Insects"

_cells, 2021, doi:10.3390/cells10020470_

Round 1

Reviewer 1 Report

The subject of the paper is appropriate for the publication in the journal Cells. The submitted paper is a review dealing mainly with regulation of cytochrome c oxidase (COX) in flying insects. Generally, the authors provide a view on the role of regulation of mitochondria and redox metabolism in flying insects and especially on the biological roles of the COX regulation (mainly through phosphorylation of COXIV and COXVI subunits of the enzyme) on the several physiological activities of flying insects. The review is written well, the paragraphs are logically ordered, and as whole provide useful information about the topic.

The fact that the allosteric regulation of COX is performed by adenylates (ATP/ADP ratio) is already well known. There exist number of articles which support this hypothesis, mostly in mammalian organisms. On the other hand, only few works deal with this topic in insects.

The content of the manuscript, however, does not fully correlate with the expression in the abstract:”This review will discuss the biology of COX in flying insects from a comparative perspective, focusing on the molecular, structural and kinetic features of this enzyme complex to distinct flight behaviors and considering the potential regulation by adenylates”. There are provided only few experimental data that describe the molecular, structural and kinetic behavior of this enzyme in flying insects.  

Further, the authors should provide more data that support their notion in conclusion “the allosteric regulation COX activity by adenylate might acts as a fine –tuning mechanism to modulate mitochondrial oxidant production in insects with potential consequences to dispersal, reproduction, immune response and longevity”. This statement is rather hypothetical and not substantiated by real data.

The possible role of ∆ψ on the regulation of COX should be also discussed.

It is not clear what the authors mean by the sentence: “Several biological and physiological aspects were associated with altered COX function in flying insects” (p.7., l. 310). How is the COX function altered in this situation?

On several sites in the manuscript it is mentioned that COX has higher activity under certain conditions compared to the other conditions. The increase of COX activity is connected only with higher COX concentration or the catalytic efficiency of COX is changed?

Regulation of COX activity emerges as a key mechanism to mediate and modulate a number of physiological processes in flying insects (p. 8)”. This notion is not confirmed by experimental data.

Discussion about the physiological role of monomer/dimer equilibrium in COX should be extended.

Only minor grammatical, stylistic, and formal errors are present in the manuscript, e.g.:

- 800Hz (p.2, l. 76), spacing is missing, it should be 800 Hz

- instead of “mitochondrial cristae membrane” it should be better to use “mitochondrial inner membrane” (p.3, l. 136)

- cyt c should be cyt c (p.3, l.141, p.8, l.390, p.10, l.474, p.11, l.508, p.14, l. 645)

- The sentence: “This strongly suggests that G3PDH is the main mechanism ….”  it should be rewritten because the enzyme (G3PDH) is not a mechanism (p.4, l.168)

- 7L (p.6, l. 234), spacing is missing, it should be 7 L

- CuA sites (p.8, l. 370), it should be CuA sites

- “Phosphorylation….represent” (p.9, l.405-406) it should be “Phosphorylation …represents…”

- 17kDa (p.9, l. 410, 411), spacing is missing, it should be 17 kDa

Recommendation: Before publication, the authors should address the above mentioned points.

Author Response

Please see attached the rebuttal letter. Thanks.

Reviewer 2 Report

Summary: In this article, Mesquita et al. review the role of cytochrome c oxidase (COX) in flying insects and its potential regulation by adenylates. The allosteric regulation of COX activity by adenylates provides relevant insights into energy metabolism regulation and in the redox balance. This review is impressive; however, some concerns should be addressed before publication.
Minor changes:

1.    Line no. 81, fix Ca+2 to Ca2+ throughout the manuscript. 
2.    In line no. 242, It would be good to add the role of mitochondrial calcium in energy metabolism regulation.
3.    Most of the references are out of date. 
4.    The authors need to define acronyms the first time they are referenced.
5.    There are many grammatical and typo errors throughout the text. For example
•    Line no 293-294, please remove (…) from the text  “...the best material for the study of the absorption spectrum of cytochrome...”. Keilin also noted that “... among
•    Line no 470, fix font size, and remove underline “some prokaryote genomes.”

Author Response

Please see attached the rebuttal letter. Thanks
